# Current Development of Nano-Drug Delivery to Target Macrophages

**DOI:** 10.3390/biomedicines10051203

**Published:** 2022-05-23

**Authors:** Donglin Cai, Wendong Gao, Zhelun Li, Yufeng Zhang, Lan Xiao, Yin Xiao

**Affiliations:** 1Centre for Biomedical Technologies, School of Mechanical, Medical & Process Engineering, Queensland University of Technology (QUT), Brisbane, QLD 4000, Australia; donglin.cai@hdr.qut.edu.au (D.C.); w24.gao@qut.edu.au (W.G.); zhelun.li@hdr.qut.edu.au (Z.L.); 2State Key Laboratory Breeding Base of Basic Science of Stomatology (Hubei MOST) and Key Laboratory of Oral Biomedicine Ministry of Education, School and Hospital of Stomatology, Wuhan University, Wuhan 430079, China; zyf@whu.edu.cn; 3Australia-China Centre for Tissue Engineering and Regenerative Medicine, Queensland University of Technology, 60 Musk Ave., Kelvin Grove, Brisbane, QLD 4059, Australia

**Keywords:** macrophages, nanoparticles, nanotechnology, drug-targeting, inflammation, inflammatory diseases

## Abstract

Macrophages are the most important innate immune cells that participate in various inflammation-related diseases. Therefore, macrophage-related pathological processes are essential targets in the diagnosis and treatment of diseases. Since nanoparticles (NPs) can be preferentially taken up by macrophages, NPs have attracted most attention for specific macrophage-targeting. In this review, the interactions between NPs and the immune system are introduced to help understand the pharmacokinetics and biodistribution of NPs in immune cells. The current design and strategy of NPs modification for specific macrophage-targeting are investigated and summarized.

## 1. Introduction

Nanoparticles (NPs) have been extensively used for drug delivery in disease treatment, taking advantage of their stability, biocompatibility, blood circulation, immunogenicity, and capability to control drug release. Due to the nature of phagocytes, NPs can be preferably taken up by phagocytes in vivo, facilitating phagocyte-targeting drug delivery without influencing the function of other cells, which has become a new direction of drug delivery. Phagocytes, such as macrophages, are the most significant innate immune cells, which participate in the pathological processes of various inflammation-related diseases, making macrophages essential targets for developing novel diagnostic imaging and disease treatment. Therefore, increasing studies have used NPs for macrophage-targeting drug delivery. In this review, we introduced the current strategies for NPs modification for specific macrophage-targeting and their applications in inflammatory diseases to provide foundations for developing/optimizing macrophage-targeting NPs in the future.

## 2. The Physiology and Pathology Function of Macrophage

### 2.1. The Role of Macrophages in Immune Response

Macrophages, which are a class of innate immune cells that originated from the mononuclear phagocyte system (MPS), play an essential role in the inflammation and regeneration of injured tissues [1,2]. Macrophages can be derived from circulating monocytes, or exist in the body as resident cells, residing mainly in the liver (Kupffer cells), lungs (alveolar macrophages), spleen, lymph nodes, thymus, gut, marrow, brain, connective tissue, and serous cavities [3,4]. In acute inflammation, the released chemokines and cytokines induce the recruitment of monocytes which then differentiate into macrophages. As a class of phagocytic cells, macrophages mainly function in the removal of pathogens, cell debris, apoptotic cells, and small particles in immune responses [2]. Macrophages also act as antigen-presenting cells, which present antigens to T cells to stimulate the adaptive immune response [2].

Based on different functions, characteristics, surface markers, and inducers, macrophages can be broadly divided into M1 (including M1a and M1b sub-populations) and M2 phenotypes (including M2a, M2b, and M2c sub-populations) [5]. The M1 activation induced by lipopolysaccharide (LPS) or cytokines such as interferon-gamma (IFN-γ) and tumor necrosis factor-alpha (TNF-α), is characterized by high antigen presentation capacity, the killing of intracellular pathogens, secretion of pro-inflammatory cytokines and anti-tumor effects. The M2 polarization induced by interleukin-4 (IL-4) and IL-13, mainly results in the resolution of inflammation and repair of injured tissue. M1 phenotype induces the expression of inflammatory cytokines such as IL-1β, IL-6, nitric oxide (NO), and inducible nitric oxide synthase (iNOS), resulting in enhanced inflammation. In contrast, the M2 phenotype is characterized by an increase in the IL-10 cytokine and a decrease in the production of iNOS, leading to limitation of inflammation and promotion of tissue repair [6,7].

### 2.2. Macrophages in Diseases with Inflammation

#### 2.2.1. Infection

Since macrophages are prominent in inflammation progress, they are closely related to the pathology of many inflammation-related diseases. In infectious diseases, macrophages contribute to the defense against pathogens and maintenance of tissue homeostasis. Pathogens can be detected by macrophages by combining with the specific receptors on the cell surface, such as Toll-like receptors (TLR) or other pathogen recognition receptors, leading to activation of downstream signaling pathways and enhanced release of cytokines to alarm other cells about bacterial invasion. Macrophages also engulf the alien microorganisms and even infected cells, degrade antigens in the early stage of an infectious process. Then, the activated macrophages will function as antigen-presenting cells and translocate degraded antigens to cell surfaces. CD4^+^ T cells can further recognize those antigens by T cell receptors (TCR) and be activated to initiate Th1 and Th2 immune responses. Activated macrophages can also release cytokines that can influence the adaptive immune response [1]. Macrophages are generally activated towards a M1 phenotype in the early stage of the infection, leading to the release of a large amount of pro-inflammatory factors (TNF-α, IL-1β, NO, etc.), thus killing the invading organisms and activating the adaptive immunity (through antigen-presenting). To avoid the excessive inflammatory response, macrophages undergo apoptosis or polarize to the anti-inflammatory M2 phenotype to protect the host from excessive injury and facilitate tissue healing.

In brief, the M1 phenotype dominates in infection and is strongly microbicidal, exhibiting high antigen presentation ability [8,9]. In contrast, M2 phenotype is less microbicidal and is associated with parasite infestation or chronic infection. It is worth noting that some bacteria can evade macrophage clearance through some special mechanisms. After being engulfed, the bacteria are supposed to either escape from the phagosome, or restrain the fusion of phagosome and lysosome, or survive inside phagolysosomes [10,11,12]. For example, *Listeria monocytogenes* can escape from phagosome [13]; bacteria from *Shigella* genus voids formation of autophagosome [14], *Mycobaterium tuberculosis* impedes phagosome maturation [15]; *Legionella pneumophila* modifies the phagosome [16]; and *Coxiella burnetti* replicates inside phagolysosomes [17].

#### 2.2.2. Bone Healing

Macrophages have also received the most attention in bone healing than other immune cells due to their vital role in osteogenesis and osteoclastogenesis. It has been reported that depletion of resident macrophages leads to the failure of osteoblast-driven bone formation in vivo, indicating that resident macrophages are crucial for osteoblast differentiation and mineralization [18]. Macrophages are also the precursor of osteoclasts. Under the stimulation of macrophage-colony stimulating factor (M-CSF) and receptor activator of nuclear factor-κB (NF-κB) ligand (RANKL), macrophages can differentiate into osteoclasts and modulate the formation of new bone. It is still controversial about which macrophage phenotype is the most beneficial for osteogenesis. Traditionally, the pro-inflammatory M1 macrophages are regarded to induce osteoclastic activities by secreting various pro-inflammatory cytokines, resulting in enhanced bone resorption. On the contrary, activated M2 macrophages tend to be more anti-inflammatory and closely associated with late-stage of tissue repair, resulting in the formation of new bone. However, some recent studies have found that M1 macrophages are also capable of enhancing osteogenesis [19,20]. Moreover, premature of M2 macrophages and excessive M2 polarization lead to formation of scar tissue or delayed wound healing [21,22]. Therefore, it is assumed that both M1 and M2 macrophages play indispensable roles in osteogenesis and a timely switch of M1 to M2 phenotype is the key factor in bone healing. The pro-inflammatory M1 macrophages play a more prominent role in the early stage of the repair response, while the M2 macrophages dominate the mid to late stages of the repair response. The M1 polarization in the early stage of bone healing determines the cytokine release pattern of M2 macrophages. Prolonged M1 polarization can lead to an increase in fibrosis-inducing cytokine release patterns (such as TNF-α, tissue growth factor-β1 (TGF-β1), TGF-β3), which results in the formation of a fibrocapsule. By contrast, an effective and timely M1-to-M2 macrophage phenotype switch can result in an osteogenesis-favoring cytokine release pattern (such as bone morphogenetic protein (BMP) and vascular endothelial growth factor (VEGF)) to facilitate the formation of new bone tissue [23].

#### 2.2.3. Cancer

Cancer refers to the rapid, abnormal, and uncontrolled proliferation of local tissue cells. It is commonly believed that tumor-associated macrophages (TAMs) assist in the angiogenesis, metastases, and immunosuppression of tumors, thus playing a prominent role in tumor survival [24,25]. TAMs express low levels of the polarization-associated macrophage antigens, carboxypeptidase M, TNF-α, and CD51, but show high expression of IL-1 and IL-6. Although TAMs can also be broadly divided into M1 and M2 phenotypes [26,27], it is believed that most resident TAMs exhibit a M2-like phenotype induced by chemokines, tumor-derived cytokines, and proteases, such as IL-4, IL-6, IL-10, TGF-β1, and polyethylene glycol (PEG) [28,29]. M2 macrophages promote tumor angiogenesis by secreting matrix metalloproteinases (MMPs) to release matrix-sequestered VEGF and by producing dozens of angiogenic factors that facilitate endothelial survival and proliferation. Enhanced angiogenesis further promotes tumor growth which is associated with faster cancer progression and shorter patient survival [4,30,31,32,33]. M2 macrophages also secrete anti-inflammatory factors, such as arginase, TGF-β, and IL-10 [34,35] to induce immunosuppression [36]. In contrast, M1 macrophages increase antigen presentation and enhance tumor cell clearance by secreting pro-inflammatory cytokines, including IL-6, IL-12, and TNF-α. Therefore, targeting and repolarizing TAMs from M2 to M1 phenotype can be a potential strategy to prevent tumor neovascularization and growth, thus being a promising therapeutic approach against cancer [37,38,39].

#### 2.2.4. Atherosclerotic Cardiovascular Disease

Atherosclerotic cardiovascular disease is a chronic inflammatory disease, which causes morbidity and mortality worldwide [40]. During the pathological process of atherosclerotic cardiovascular disease, macrophages play a prominent role. Monocytes are recruited into the subendothelial space of the arterial intima by endothelial cells [41,42] and then differentiate into inflammatory macrophages, which phagocytose lipoproteins, leading to the formation of lipid-rich foam cells and early atherosclerotic lesions, resulting in the amplification of the local inflammatory response [43]. The atherosclerotic lesions progress to an advanced stage if the pro-inflammatory status persists, characterized by accumulated apoptotic macrophages and failure of their clearance [44,45,46]. Depending on local inducers in the micro-environment, macrophages can either polarize into a pro-inflammatory M1 phenotype or a reparative M2 phenotype [47]. Pro-inflammatory M1 macrophages are known to diminish lesion stability by inhibiting collagen production from smooth muscle cells and producing MMPs that degrade the protective fibrous capsule [48,49,50,51]. By contrast, pro-resolving M2 macrophages promote lesion stability by clearing apoptotic cells through efferocytosis, thereby diminishing plaque necrosis [52]. Therefore, macrophages are important targets in the treatment of atherosclerotic cardiovascular disease, considering their key role in the progression and regression of atherosclerotic lesions. Moreover, the so-called vulnerable atherosclerotic plaques lead to fatal sequelae, such as acute myocardial infarction and stroke, causing death worldwide. Given that atherogenesis processes slowly and sometimes over decades, early diagnosis and interventional therapy against high-risk atherosclerotic plaques before clinical symptoms are of great importance [53], and macrophages have been regarded as ideal targets of diagnostic imaging and treatment for atherosclerosis [54,55,56,57].

#### 2.2.5. Ageing

Ageing populations are becoming a global trend, characterized by the accumulation of cells which have undergone the process of permanent cell cycle arrest, termed senescence [58,59]. In the immune system, cellular senescence has been observed in multiple cell types, including macrophages and T cells [60,61,62]. Macrophage senescence has been found to contribute to ageing, evidenced by the fact that in ageing individuals, multiple organs are under a long-term, chronic, and sustainable state of inflammation, also known as inflammageing [63]. Although macrophages are not the sole source of inflammageing, extensive studies have shown that macrophages are the central component in initiating ageing-related chronic inflammation. It has been found that the M1 macrophage level in ageing mice is higher than that in young mice, which is probably related to deficient tissue repair capacity and long-term inflammation state in ageing mice [64]. Meanwhile, the regenerative M2 macrophage population decreases with age, symbolized by the reduced expression of IL-10 (a M2 polarization marker) in ageing mice [65], and delaying macrophage senescence by knocking out tumor suppressor gene *p16* (*INK4a*), which is associated with an increased risk of age-related inflammatory diseases, can upregulate the expression of Arg-1 (a M2 polarization marker) [66]. In addition to phenotype switch, the total macrophage population also changes with ageing. Studies have compared bone marrow in teenagers (<19 years of age) with adults, and found the former contains significantly more macrophages compared to the latter [67]. These macrophages have been further analyzed, and researchers have identified a higher population of CX3CR1+ macrophages and a reduction in Ly6C+ macrophages in old mice compared to young [68], suggesting that macrophages changed towards a more pro-angiogenic phenotype in old mice compared with young mice [69]. In the lungs of old mice, studies found the macrophage numbers declined, with the expression of over 3000 genes being altered, as compared to lungs of young mice [70]. Among these genes, the most affected ones include scavenging receptors, such as CD204 [70], and the macrophage receptor with collagenous domain (MARCO) [71], which impacts bacterial phagocytosis and efferocytosis of apoptotic neutrophils, adversely affecting inflammatory responses.

Macrophages also contribute to the pathology of age-related diseases, such as polymyalgia rheumatic (PMR) and giant cell arteritis (GCA), which only occur in people over 50 years old and are both characterized by increasing production of IL-1β and IL-6 by arterial macrophages [68]. Likewise, chronic obstructive pulmonary disease (COPD) is also strongly linked to inflammageing [72,73,74], due to the impaired bacterial phagocytosis and efferocytosis in alveolar macrophages, resulting in increased bacterial colonization and an elevated pro-inflammatory environment [75]. Therefore, it is believed that inflammation caused by aged macrophages has been potential target to reverse the detrimental effects of inflammageing and boost immunity in old individuals.

Considering the prominent role of macrophages in the pathological progression of various diseases, they are regarded as essential targets in modulating inflammation, thus developing new methods for disease treatment. However, specifically targeting macrophages involved in conditions without interfering with host immunity remains a challenge [76]. Given the phagocytic nature of macrophages in the clearance of cellular debris, apoptotic cells, pathogens, and especially foreign particles in immune response, NPshave attracted much attention for specific macrophage-targeting due to their feature to be preferentially engulfed by macrophages [2].

## 3. Nanotechnology-Based Drug Delivery System

NPs, with diameters ranging from 1 to 500 nm, have high loading capacities for drugs owing to their large surface-area-to-volume ratios. The small size of NPs allows them to be largely endocytosed by cells, thus increasing their accumulation in phagocytes [2]. Their small size also increases the tumor-targeting ability of NPs through the enhanced capillary permeability and retention effect as tumor tissues lack lymphatic drainage compared with normal tissues, which enables NPs to be taken up more efficiently by tumors [77]. NPs can be synthesized into various shapes and sizes based on different preparation methods with different raw materials (such as polymers, lipids, ceramics, and metals). Because of their tunable physiochemical properties and enhanced carrying capacity, NPs have been used for various biomedical applications [78,79,80], including diagnostic imaging and drug delivery. It is well known that drug-loaded nanosystems have many advantages compared with traditional drug therapy in terms of stability, biocompatibility, controlled drug release, and immunogenicity. Studies have shown that the encapsulation of therapeutics (such as small-molecule drugs, peptides, proteins, small interfering RNA (siRNA), and microRNA (miRNA)) into NPs prevents extensive metabolism and prolongs circulation time, improving the chemical stability and pharmacokinetics of the loaded therapeutics [81,82]. Moreover, NPs are known to interact with immune components in blood, such as opsonin, which causes NPs to be more easily recognized and preferentially taken up by macrophages [83,84], thus reducing the off-target and side effects on other cells and tissues compared with free drugs [85,86,87]. Until now, NP-based delivery strategies have shown great promise in suppressing specific pathological processes, including inflammation in mouse models [81].

NPs can be assigned into two major groups: inorganic (e.g., gold, titanium oxide, cerium oxide, iron oxide, and quantum dot (QD) NPs) and organic (e.g., liposomes, micelles, protein-based nanomaterials, and dendrimers) (Figure 1).

Inorganic NPs are normally much smaller, contain metallic elements, and exhibit physical properties of specific nanoscale [88]. For example, gold NPs are extensively studied in cardiac tissue regeneration due to their capability to enhance scaffold conductivity [89]. Moreover, recent research reported their ability for immunomodulation by restraining inflammatory cytokines and enhancing anti-inflammatory cytokines [90]. The surface chemistry of gold NPs can be changed by bioactive molecules, allowing them to modulate macrophage polarization. Studies have decorated gold NPs with peptide arginine-glycine-aspartic acid (RGD) [91], hexapeptides Cys-Leu-Pro-Phe-Phe-Asp [92], and IL-4 [93], and those functioned gold NPs increased M2 macrophage polarization and decreased M1 phenotype. Titanium oxide NPs (TiO_2_ NPs), which are largely used in tissue engineering due to their ability to resist corrosion from the body, were also reported to accelerate M1 to M2 transition [94]. Research by Lee has reported an increased M2 macrophage population on calcium (Ca) and strontium (Sr)-modified titanium implants, leading to enhanced osteogenesis [95]. Moreover, it has been reported that TiO_2_ nanotubes can induce anti-inflammatory phenotype and promote endothelialization [96]. Cerium oxide NPs (CeO_2_ NPs) are also bioactive NPs with strong antioxidant properties. CeO_2_ NPs have been reported to enhance wound healing by enhancing fibroblast and vascular endothelial cells [97]. Moreover, by coating hydroxyapatite on the surface of CeO_2_ NPs, an increased M2 polarization can be achieved [98]. The Ce^4+^/Ce^3+^ ratio can influence the osteogenic ability of CeO_2_. The higher ratio enhanced M2 macrophages polarization and achieved better osseointegration in vivo [99]. Despite immunomodulation, inorganic NPs are also applied as imaging agents because of their superparamagnetic properties [100,101]. For example, both gold, iron oxide and QD NPs can be used in magnetic resonance imaging (MRI). Since the optical and electronic characteristics of QD NPs directly depend on the particle size, QD NPs can be applied in various imaging techniques other than MRI, such as X-ray computed tomography (CT), positron emission tomography (PET), and single-photon emission computed tomography (SPECT), by simply changing the QD NPs size [102,103,104,105].

Organic nanocarriers are carbon-based and are generally characterized by their biocompatibility and improved drug-loading capacity. Several major groups of organic nanocarriers include liposomes, micelles, dendrimers, and protein/peptide-based nanocarriers [88]. Liposomes are large NPs (25–1000 nm) that have been shown to accumulate in macrophages (by inducing phagocytosis), achieving macrophage-targeting in a simple and effective way [106]. However, due to their enhanced macrophage uptake in the mononuclear phagocyte system (MPS), liposomes can be rapidly cleared from circulation. Some studies coated PEG on the surface of liposomes and reduced their clearance by the MPS [107]. Micelles are natural nanocarriers with a large water-free environment in their core, which is surrounded by a hydrophilic surface. Micelles can be functioned by different bioactive molecules, such as CpG oligonucleotides and N-methylpyrrolidone. CpG oligonucleotides enable micelles to increase the CD8+ and CD4+ T cell responses, and N-methylpyrrolidone reduces the accumulation of micelles in the reticuloendothelial system (RES) [108,109]. Dendrimers are tree-like nanostructures, which have attracted significant attention as nanocarriers for small-molecular drugs, proteins, and nucleic acids. The subunits of dendrimers grow around the core in concentric layers to produce stepwise increases in size that are similar to globular proteins. This property is beneficial to NP–drug interaction [110].

## 4. Interaction between NPs and Immune System

The clearance and biodistribution of NPs are mediated by circulating and tissue-resident phagocytes of the MPS, including macrophages, which engulf and accumulate NPs in tissues [111,112]. This interaction between NPs and the immune cells can either result in immunosuppression or in immunostimulation, which may enhance or reduce the therapeutic effects of NPs [84,113]. Therefore, understanding the mechanisms involved in NP-uptake and the interaction between NP and MPS can facilitate the development of advanced NPs with features, such as prolonged systemic circulation time, enhanced selective delivery to the targeted tissues, as well as improved clinical implication.

Once entered the blood circulation system, plasma proteins will quickly adhere to the surface of NPs to form a “corona”, which contains about 3700 proteins [114]. Among these proteins, the adsorption of opsonin on the NP surfaces enables NPs to be more engulfed by phagocytes in the RES system, especially by macrophages [115]. The NP-uptake in macrophages is either non-selective or selective. The nonselective mechanisms include macropinocytosis and phagocytosis. Macropinocytosis refers to the uptake of tiny, soluble substances or extracellular fluids, while phagocytosis involves the uptake of larger, solid particles [116]. Previously mentioned opsonin can enhance phagocytosis by binding to macrophage surface receptors once adsorbed on the NP surface to trigger phagocytosis. Local actin will then be rearranged, contributing to the formation of a new vesicle called “phagosome” once tightly engulfed large NPs or NP aggregates. The fusion of phagosome with lysosome further forms a “phagolysosome”, which will eventually be degraded [117]. Unlike nonselective uptake, selective uptake requires the interaction and combination of cell membrane receptors with ligands decorated on NP surface [118,119,120]. The most common form of selective uptake is clathrin-mediated endocytosis, which strongly depends on NP size and only occurs in NPs with a diameter less than 50 nm [121,122]. As a coating protein presents in all cells, clathrin can be activated by the binding of ligands on the NPs to the AP-2 adaptor complex on the cell membrane, and then move to the binding site where a transport vesicle is formed. The vesicle will be coated with clathrin to travel through the cell. After vesicles are fused with lysosomes, the clathrin will fall off and return to the cell surface to repeat the process [123].

Many characteristics can influence the interaction between NPs and macrophages. For example, greater size may increase the extent of opsonization and enhance NP-uptake, and smaller size NPs (<100 nm) are more preferable due to slower removal from blood [124,125]. In addition to NP size, surface properties, such as hydrophilicity, and surface charge can also impact the adsorption of opsonin, thus influencing the uptake of NPs by phagocytes and clearance from the bloodstream [126,127]. Increased hydrophilicity results in a lower degree of protein adsorption and reduced uptake by macrophages [128]. Tabata et al. have tested the uptake of hydrophilic and hydrophobic NPs by macrophages and showed that the latter were more largely engulfed [129]. Many researchers coated NPs with PEG to reduce their uptake by macrophages and prolong the circulation lifetime. PEG is a hydrophilic polymer that increases the hydrophilicity of NPs and provides a steric hindrance via water shell formation, thus preventing the opsonin from binding to the NPs [128,130]. Purified cellular membranes from leukocytes and platelets are also used to avoid opsonization and delay the uptake of NPs by MPS [131,132]. Surface charge is another factor that influences phagocytosis. Positively or negatively charged surfaces show increased phagocytosis, as compared to neutral particles [120,133], which is resulted from the smaller charge magnitude of neutral particles [133]. Smaller aspect ratios also allow NPs to be phagocytosed more easily by macrophages compared with NPs with larger aspect ratios, due to the fact that smaller aspect ratios require less actin remodeling and less energy in the phagocytic process [134].

The uptake of NPs by phagocytes also has toxicological effects on immune cells [82], which might be contributed by the particle chemistry of NPs, according to some studies [135,136,137]. For example, the surface charge of silicon (Si) NPs and the size of zinc oxide (ZnO) NPs affected the interaction with cells and intracellular reactive oxygen species (ROS) production, thus influencing cytotoxicity [136,137]. Compared with T and B lymphocytes, macrophages play a more critical role in the biodistribution of NPs, which may be more sensitive to NP toxicity [138]. This indicates that NP-uptake by macrophages and intracellular release of the loaded drugs may contribute to the NP-mediated toxic effect on the MPS cells.

## 5. Macrophage-Targeting Nanotechnologies

### 5.1. Passive Macrophage-Targeting Nanotechnologies

There are two specific-targeting approaches in the design of NP-based drug delivery systems, namely passive and active targeting (Figure 2).

Passive targeting takes advantage of the NP pharmacokinetics, unique vascular pathophysiology, and immune responses of the targeted tissue, leading to the accumulation of NPs [139,140]. For example, large NPs (up to 500 nm) predominantly accumulate in the liver and lungs; medium NPs (10–300 nm) aggregate mainly in the liver and spleen after being opsonized and removed from the circulation; and small NPs (1–20 nm) are usually degraded by macrophages in the kidneys [139,141,142]. Therefore, the preference for unmodified NPs to accumulate in certain tissues has been utilized as a passive-targeting approach to deliver the payload to the macrophages. Inflammatory tissues and solid tumors are characterized by vascular leakage contributed by inflammatory mediators, cytokines, and growth factors that cause disruption of the endothelium [143,144,145] and, consequently, result in NP in situ accumulation. For example, Corvo et al. [146] intravenously injected liposomes coated with PEG into a mice model of rheumatoid arthritis and observed passive accumulation of these NPs at the arthritic sites. Thus, NPs with proper size can preferentially extravasate from the blood into the interstitial spaces and accumulate in inflammation sites or tumor tissues via the enhanced permeability and retention (EPR) effect [147]. Meanwhile, Keliher et al. [148] reported that TAMs are able to selectively capture NPs and translocate them from the periphery to the central hypoxic zone of tumor tissue. The ability of TAMs to take up NPs has been utilized in tumor imaging and quantification of TAMs in which zirconium (Zr)-labeled dextran-based NPs were applied as MRI imaging agents [142]. Since elevated infiltration of macrophages and lymphocytes are common in the extravascular spaces of inflammation sites and tumors (inducing excessive inflammatory responses and tissue damage) [149,150,151], macrophages are considered the primary therapeutic targets in inflammatory diseases [152,153], and particle uptake into macrophages could, therefore, allow the selective accumulation of NPs in the areas of inflammation [154,155].

### 5.2. Active Macrophage-Targeting Nanotechnologies

#### 5.2.1. Phagocytosis-Related Cell Membrane Receptors Targeting

Active targeting can be used to enhance the selective delivery of drugs into the target cells or tissues by exploiting the specific interactions between drug carriers and targeted sites [139,156,157,158]. Currently, the most common approach to achieve active-targeting is to decorate the surface of NPs with an agent (e.g., ligand, antibody, and peptide) that can selectively interact and be recognized by the particular cell type in certain tissue [159]. Different receptors and lipid components on the cell membrane, thus playing a prominent role in active-targeting, for they can recognize specific agents on NPs. Moreover, compared with normal cells, pathological cells may uniquely express these receptors or express them at a different amount. Depending on the target cell and/or tissue types, various ligands, such as monoclonal antibodies, could be harnessed to the particle surface [160]. Thus, the active-targeting approach of NPs is based on the concept that the surface of the delivery vehicle is modified with a ligand or antigen to allow selective interaction with the target receptors [161]. The macrophage cell membrane surfaces contain many receptors that determine the activity of macrophages, including growth, differentiation, activation, recognition, endocytosis, migration, and secretion [162]. Among them, three main groups are frequently assumed: TLR, non-TLR, and opsonic receptors. Although TLRs do not participate in phagocytosis or endocytosis, they are involved in antimicrobial peptide production and innate immunity, playing a central role in recognition of pathogens and activation of innate immune responses [163,164,165]. Non-TLR, including the family of scavenger receptors [166] and the C-type lectins [167], are involved in phagocytosis and endocytosis. Opsonic receptors include complement receptors (integrins, such as CD11b) and Fc receptors (immunoglobulin superfamily, such as CD44), which dominate the phagocytosis and endocytosis of complement- or antibody-opsonised particles, respectively [168]. Those phagocytosis-related receptors are, thus, ideal structures for a macrophage-targeting therapy, which can facilitate nano-carriers to deliver therapeutic agents into macrophages selectively. As for non-TLR, for example, C-type lectin receptors (CLRs) recognize conserved carbohydrate structures, including mannose and galactose. Mannosylated liposomes have repeatedly been shown to preferentially target macrophages, enhancing cellular uptake both in vitro and in vivo [169,170,171,172]. Likewise, Lipinski et al. have decorated antibodies that specially interact with scavenger receptor (CD36, which is highly expressed on the surface of macrophages) on the surface of micelles for macrophage-targeting imaging in atherosclerosis [173]. Hyaluronic acid (HA), which is an unbranched non-sulphated glycosaminoglycan consisting of repeating disaccharides (β-1,3-N-acetyl-D-glucosamine and β-1,4-D-glucuronic acid), can be recognized by a large number of HA receptors on macrophages, such as Fc receptor (CD44), the receptor for HA-mediated motility (RHAMM), and several other receptors possessing HA-binding motifs [174]. Kamat et al. found that iron oxide magnetic NPs coated HA could be efficiently entrapped by activated THP-1 macrophages in vivo [175]. Tsai et al. reported a HA conjugated gold nanorod (HA-Au NR) as a drug carrier for anticancer doxorubicin (Dox) delivery, and HA significantly improved the recognition and uptake of HA/Dox-Au NRs by RAW 264.7 cells through the specific interaction with the HA receptor, CD44 on the cell surface [176]. Additionally, the modification effectively improved biocompatibility by switching the surface charge from positive (due to chitosan polymer) to negative (due to hyaluronate) [1]. To reduce the immunogenicity associated with the Fc portion, Gagne et al. coupled Fab’ fragment of the anti-HLA-DR (anti-class II major histocompatibility complex molecules) antibody rather than the entire antibody with PEG-modified immunoliposomes. Macrophages express a high level of HLA-DR, and the result showed a much more increased accumulation of modified NPs in lymphoid tissues compared with free drugs [177].

#### 5.2.2. Pathogen Components-Mediated Macrophage-Targeting

In addition to components targeting phagocytosis-related cell membrane receptors, another strategy is to use pathogen-associated components to induce active-targeting, which takes advantage of the macrophage’s nature to engulf pathogens. Chavez-Santoscoy et al. decorated galactose and di-mannose on the surface of polyanhydride NPs to provide pathogen-like properties to target CLRs on alveolar macrophages [178]. Mannosylated fluorescent phenylboronic acid-containing NPs were also selectively taken up by murine RAW 264.7 macrophages, which were used as cell imaging agents [179]. More importantly, according to some in vivo studies, mannose significantly increased the uptake of gelatin NPs by macrophages in the liver, lymph nodes, and lung, compared with pure gelatin NPs [1]. Dextran sulphate, derived from certain *lactic acid bacteria*, is constituted by a linear glucose chain with one sulphate group per glucose unit [180]. Dextran is reported to be recognized by scavenger receptor class A on macrophages [181,182,183,184], which is, therefore, frequently used as decoration on NPs to enhance the specificity for targeting macrophages [148,185]. Keliher et al. has introduced a crosslinked, short-chain dextran nanoparticle that accumulated primarily in tissue-resident macrophages. Labeled with ^89^Zr, this NP can be used as a macrophage-specific PET imaging agent to quantify macrophage inflammation levels in the diagnosis of various diseases, such as cancer, atherosclerosis, and myocardial infarction [148].

Despite of ligands, Ren et al. encapsuled polymer−lipid hybrid NPs into porous and hollow yeast cell wall for macrophage-targeting drug delivery. The yeast cell wall composed of natural β-1,3-D glucan, can be recognized by the apical membrane receptor, dectin-1, which has a high expression on macrophages and intestinal M cells [186]. Soto et al. also incorporated NPs as cores inside glucan particles (GPs), which are hollow, porous 2–4 μm microspheres derived from the cell walls of Baker’s yeast (*Saccharomyces cerevisiae*), taking advantage of the macrophage-targeting property of GPs. The 1,3-β-glucan outer shell provides for receptor-mediated uptake by phagocytic cells expressing β-glucan receptors [187]. Except for the cell wall, Gram-negative bacteria outer membrane vehicles (OMVs), which exhibit various pathogen-associated molecular patterns and immunogenic antigens [188,189] that can be recognized by macrophages, have also been used for macrophage-targeting immunomodulation [190,191]. Gao et al. compared nanoparticles coated with the membrane of OMVs from *Staphyloccocus. aureus* (*S. aureus*) with counterparts coated with PEGylated lipid bilayer, and OMV membrane coating was found to facilitate NP internalization by *S. aureus*-infected macrophages [192].

Despite membrane coating, bio-nanocapsules (BNCs) derived from pathogens could be directly used as drug carriers. For example, hepatitis B virus (HBV) envelope particles, which are 50 nm BNCs consisting of approximately 110 molecules of HBV surface antigen L protein and lipid bilayer, have been explored to specifically deliver payloads to liver cells [193]. Li et al. then developed mutated BNC to selectively target the non-hepatic cells and tissues in vitro and in vivo, relying on the outwardly displayed tandem form of the *S. aureus* protein A-derived Z domain which could bind to animal IgG Fc domains [194,195,196]. This protein A-derived Z domain was recently replaced by *Finegoldia magna* protein L-derived B1 domain and showed enhanced uptake by the murine macrophage cell line RAW 264.7 [197].

#### 5.2.3. Other Chemical Compounds-Mediated Macrophage-Targeting

The folate receptor (FR) is over-expressed on the activated macrophage surface in rheumatoid arthritis [198,199]. Thomas et al. used folate, which can be recognized by FR on macrophages, to decorate NPs loaded with methotrexate for macrophages-targeting. These NPs were demonstrated to offer a practical clinical approach to improving the drug delivery and efficacy at the inflammatory site of arthritis [200]. Similarly, another study fabricated FR-targeting fluorescence nanoprobes to detect and quantify the extent of biomaterial-mediated inflammatory responses in vivo. They found a good relationship between the extent of the inflammation and the intensity of nanoprobe-associated fluorescence signal in tissue [201]. Except for arthritis, ovarian TAMs also express a high level of folate receptor-2 (FOLR2), which can be selectively targeted using G5-methotrexate (act as both a ligand and a toxin) dendrimer NPs for cancer treatment [202].

Phosphatidylserine (PhoS) has significant potential to selectively target macrophages and has been frequently applied in developing drug delivery systems by anchoring PhoS on NPs [200,201,202,203]. Normal cells have PhoS inside the phospholipid bilayered plasma membrane, whereas apoptotic cells bring out PhoS to the outer surface of the plasma membrane and make themselves recognized by macrophages for phagocytosis. Thus, the expression of PhoS on apoptotic cells allows for PhoS-dependent identification and engulfment by macrophages [203].

Brain-derived neurotrophic factor (BDNF), which is a protein belonging to neurotrophins that support neuronal cells’ growth, differentiation, and survival, can bind to two neurotrophin receptors on the cell surface: the low-affinity neurotrophin receptor p75 and the high-affinity receptor TrkB [204,205]. In the research by Talvitie, chitosan NPs were functioned by TrkB binding targeting peptides and showed more efficient binding to RAW 264.7 macrophages than pure NPs [206].

Heparin, which is traditionally regarded as an anticoagulant, is also reported to have macrophage binding affinity and can further act as a macrophage-targeting agent [207]. Its uptake by macrophages is reported to be mediated by scavenger-like receptors. Interestingly, heparin-loading reduced the toxicity of cationic NPs in the rat macrophage NR8383 cell line [208,209].

#### 5.2.4. Strategy to Specifically Target M1 or M2 Macrophage

As previously mentioned, macrophages in inflammation sites polarize into two phenotypes depending on local stimulations. A prolonged activation or dysregulation of M1 activity is closely related to the development of chronic diseases, such as rheumatoid arthritis, delayed/non-healing wounds, psoriasis, and septic shock, which can lead to multiple organ dysfunction syndromes (MODS). Therefore, selective M1 macrophage-targeting immunomodulation has become the focus of treatment, which can avoid the side effects on other cells [210]. However, it is challenging to find a suitable surface molecule specifically expressed or upregulated by M1 macrophages. This is because macrophages are highly plastic cells, exhibiting different surface markers depending on the inducers from the different microenvironments [27]. Recently, M1-specific and M2-specific surface marker screening have been performed, using mice and human macrophages exposed to their respective inflammatory stimulus (M1: IFN-γ and LPS, M2: IL-4) [211]. According to this study, the expression of two receptors (CD64, CD14) was increased in both mice and human M1 macrophages but reduced in the M2 macrophages. On the contrary, mannose receptor (CD206) and macrophage galactose-type C-type lectin (CD301) were down-regulated on M1 macrophages but upregulated on the M2 macrophages in both mice and human M1 macrophages [211]. Among them, attention has been paid to Fc γ RI (commonly referred to as CD64), which is considered a suitable target molecule on M1 macrophages owing to their ability to bind and rapidly internalize monomeric IgG [212]. The development of antibodies against CD64, such as monoclonal antibody (mAb) 197, can specifically recognize and bind to monocytes, allowing for the development of macrophage-targeting technologies [213]. mAb 197 was used for the clinical treatment of chronic immune thrombocytopenia purpura (cITP), as mAb 197 binding prevented CD64 mediated destruction of IgG-coated platelets [214]. The murine-derived mAb 197 was further humanized (H22), which contained both binding specificity and high affinity for CD64, to reduce immunogenic response in human body [215,216].

TAMs, which are normally M2-like macrophages, facilitate tumor angiogenesis and growth, thus playing an indispensable role in tumor development and progression. Therefore, specific M2 TAMs have been considered as therapeutic targets in tumor treatment [217]. Recently, a M2 macrophage-binding peptide (M2pep) identified by phage display was reported to have high selectivity and efficient targeting ability to M2 macrophages [218,219]. Research has functioned gold NPs with M2pep to deliver siRNA in a lung cancer mouse model to achieve specific and long-lasting gene therapy in inflammatory TAMs [218,219]. Other cancer treatments, such as NPs-based magnetic hyperthermia therapy (MHT), were also performed. By coating iron oxide NP with M2pep in an orthotopic breast cancer mouse model, the M2 TAM-targeting MHT significantly reduced the tumor volume by reducing the population of pro-tumoral M2 TAMs in tumor [217]. Except for M2pep, another α-peptide (a scavenger receptor B type 1 targeting peptide), also possesses great specificity to M2-like TAMs [219,220]. Han et al. developed poly (lactic-co-glycolic acid) (PLGA) NPs conjugated both M2pep and α-pep to target M2 TAMs. This successfully transformed M2 to M1 phenotype, and remodeled the tumor microenvironment to allow the killing of tumor cells [221].

## 6. Application of Macrophage-Targeting Nanotechnology in Disease Treatment

Given the important roles of macrophages in the progression of many diseases, macrophage-targeting therapeutics were also extensively studied in recent years. In this section, the latest developments in disease diagnosis and treatment based on macrophage-targeting NPs will be reviewed and discussed.

### 6.1. Macrophage-Targeting NPs for Diagnosis

The role of NPs as an imaging/contrast agent enhancer in noninvasive diagnostic technique has been explored due to their unique physical and chemical properties. Moreover, the NP surfaces could be modified to improve the signal intensity and specificity, allowing them to be suitable for the diagnosis of different diseases. As macrophages are involved in the pathology of many diseases (e.g., infection, tumor, atherosclerosis, and rheumatoid arthritis), this cell type has been considered as a suitable imaging target. Monitoring the role of macrophages in inflammation can help us learn more about the pathological process of inflammatory diseases [1].

The superparamagnetic iron oxide (SPIO) NPs coated with macrophage-targeting agents, such as dextran [222,223,224], human ferritin protein cage [225], osteopontin (OPN) [226] or annexin V [227], can induce signal loss in T2-weighted images and, therefore, have been applied as MRI agent to offer pivotal insights into plaque biology, thus assessing inflammatory burdens in atherosclerosis. Lipinski et al. described a scavenger receptor-targeted micelle-based nanoparticle-containing MRI contrast agent for imaging macrophages in atherosclerosis. They reported that the NP could especially target and accumulate in macrophages, which significantly increased the contrast-to-noise ratio (CNR) by 52.5%, while the untargeted NP only increased CNR by 18.7% [173]. Likewise, Li developed class AI scavenger receptor-targeting, glutathione-biomineralized gadolinium-based NPs for noninvasive precise MRI imaging to detect foam macrophages in atherosclerosis plaque. The imaging contrast showed an amplified T1 signal, which precisely targeted macrophages, and exhibited systematic clearance capabilities [228].

In addition to MRI, CT imaging, which has high spatial resolution and short acquisition time, is another frequently used technology for diagnosis. However, the low sensitivity of CT requires the administration of a high dose of NPs, which needs to be investigated in future research. For example, iodine-containing NPs have shown advantages in identifying pro-inflammatory macrophages in vulnerable plaques, but the dose of iodine-NP is 100 mg/kg for mice [229], and the potential toxicity needs to be further studied. PET, which has high tissue penetration and superior sensitivity, is another non-invasive imaging technique. Keliher et al. introduced dextran-coated iron oxide NPs labeled with ^89^Zr as macrophage-specific PET imaging agents due to their high affinity with macrophages [148]. Their later study described a PET radioactive tracer (^18^F)-labeled polyglucose NPs, which can be taken up by macrophages with high efficiency to visualize atherosclerotic plaques. This polyglucose NPs showed facilitated imaging in mouse and rabbit atherosclerosis models [230].

### 6.2. Macrophage-Targeting NPs for Tumor Treatment

As mentioned in this review, TAMs contribute to tumor development via degradation of tumor extracellular matrix, destruction of the basement membrane, promotion of angiogenesis, and recruitment of immunosuppressive cells [221]. Therefore, TAMs have been proposed as therapeutic targets. Strategies have been made to clear TAMs in the tumor environment. Miselis et al. used clodronate-containing liposomes labeled with fluorescent dye to selectively target macrophages in the tumor spheroids and eliminate these cells from the tumor environment. In the animal tumor model, the treatment resulted in a 4-fold decrease in tumor number and a 15-fold decrease in tumor size compared to the control, suggesting that the liposomes successfully decreased the tumor cell density and enhanced apoptosis of tumor cells [231]. Soto et al. used mesoporous silica NPs (MSNs) to load Dox as chemotherapeutics and resulted in enhanced tumor growth arrest. The macrophage-targeting was achieved by encapsulating MSN-Dox into GPs derived from the cell walls of *Baker’s yeast*. The outer shells of GPs express β-glucan receptors (1,3-D-glucan polysaccharide) that allow for receptor-mediated selective uptake by macrophages [187]. Likewise, Ren encapsuled polymer−lipid hybrid NPs into GPs for macrophage-targeting delivery of cabazitaxel, which showed a slower in vitro drug release and higher drug stability compared with non-coated NPs [186].

The M2-like TAMs can be reprogrammed to M1-like macrophages to induce tumor cell necrosis by secreting inflammatory cytokines. Therefore, another strategy in tumor treatment is to induce the M2-to-M1 TAM phenotype switch to remodel the tumor microenvironment [221]. According to previous studies, TLR agonists (e.g., CpG oligodeoxynucleotides (CpG ODNs), which can induce anti-tumor M1 polarization) and baicalin (which can increase the production of IFN-γ) were used as immune modulators for cancer treatment [232]. However, due to the lack of effective delivery approaches, their applications in vivo are limited, suggesting the potential of M2-targeting NPs in delivery of TLR agonists. In Han’s research, PLGA NPs conjugated M2pep and α-pep peptides were used to transform the M2-like TAMs into the M1-like phenotype by specifically delivering a combination of CpG ODN and baicalin. The result showed that the NPs were effectively ingested by M2-like TAMs both in vitro and in vivo, and the released biomolecules effectively reversed the macrophage phenotype [221]. Shan et al. also developed human ferritin heavy chain (rHF) nanocages modified with M2pep on their surfaces for the targeted delivery of CpG ODNs to M2-like TAMs. These NPs were found to inhibit tumor growth in tumor-bearing mice after intravenous injection by transforming M2 TAMs to the anti-tumor M1 type. Moreover, they discovered that the empty M2pep-rHF NPs without CpG ODNs also exhibited anti-tumor ability [233]. Opanasopit et al. [234] introduced a mannosylated liposome containing an immunomodulator called muramyl dipeptide (MDP), which is a component derived from bacterial cell wall, to target macrophages and activate the M1 polarization in an experimental liver metastases animal model. The result showed that after stimulated by MDP, the production of prostaglandins, collagenase and super-oxide anions by macrophages increased significantly, which induced cytolytic activity against tumor cells [235].

Despite immunomodulation, TAMs are also therapeutic targets for anti-angiogenic treatment. TAMs secret MMPs to release matrix-sequestered VEGF and produce dozens of angiogenic factors to facilitate endothelial survival and proliferation, thus promoting angiogenesis, as well as tumor growth [30,31,32,236]. Penn invented dendrimer NPs conjugated with methotrexate (G5-MTX NPs), a chemotherapeutic that can be recognized by highly expressed FOLR2 on the surface of ovarian TAMs, for anti-angiogenic therapy. G5-MTX NPs overcame the resistance to anti-angiogenic therapy and prevented the side-effects of anti-angiogenic therapy (which induced the generation of cancer stem-like cells) and depleted TAMs in both solid tumor and ascites models of ovarian cancer [202].

Hyperthermia therapy has attracted increasing attention for tumor treatment, as tumor tissues are particularly vulnerable to hyperthermia compared with normal tissues owing to their faster cell proliferation, enhanced hypoxia, low pH, and insufficient temperature regulation ability [237,238]. Chen et al. developed HA conjugated gold nanorods with Dox and acid-labile hydrazone linker attached to the surface as photothermal NPs. After near infrared (NIR) laser irradiation, they enhanced drug release and increased drug toxicity on tumor cells owing to the photothermal effect [176]. Similarly, Wang developed a M2 TAM-targeting iron oxide nanoparticle for MRI-guided MHT of breast tumors. MHT can penetrate deeply into tissues to treat deep tumors and magnetic NPs are commonly used hyperthermia agents. Wang’s NPs also served as contrast agents for MRI, providing diagnostic information and visualizing their distribution in vivo to guide the optimal therapeutic time window [217].

Radiation-induced fibrosis (RIF) is a dose-limiting complication of cancer radiotherapy and causes serious problems, such as restricted tissue flexibility, pain, ulceration, or necrosis [239]. The recruitment of macrophages in inflamed sites can promote inflammatory events and result in fibrosis. Therefore, macrophages are potential cellular targets for anti-inflammatory treatment by inhibiting their production of cytokines. A study successfully treated RIF in a mouse model by intraperitoneal administration of chitosan NPs carrying siRNA to silence TNF-α in local macrophage populations, which takes advantage of the natural homing potential of macrophages to inflammatory sites. They observed the uptake of fluorescently labeled siRNA NPs by peritoneal macrophages and their subsequent migration to lesion region in radiation-induced inflammatory skin, suggesting the chitosan-siRNA NPs may serve as a general therapeutic approach for inflammatory diseases [239].

### 6.3. Macrophage-Targeting NPs for Infection Control

Infectious diseases are an important health concern, as several pathogens have adapted to survive inside the phagocytic cells, especially macrophages. In some cases, macrophages even serve as nutrient reservoirs to facilitate pathogen growth and spread. Therefore, new therapeutic strategies should be developed to allow for macrophage-targeting drug delivery [162].

*Mycobacterium tuberculosis* (*Mtb*) is one of the most threatening pathogens for its latent infection in macrophages. The intracellular *Mtb* isolated itself from drugs and could spread via macrophages [172]. *Mtb* displays lipoarabinomannan (LAM) on their cell wall. LAM contains mannose oligosaccharides at the terminus of the molecule, which dominate the attachment of bacteria to macrophages [15,240,241]. Therefore, mannosylated carriers can specifically target and competitively bind to macrophages to deliver antitubercular drugs. A mannose-modified macrophage-targeting solid lipid NP has been developed to load the pH-sensitive prodrug of isoniazid (INH) to treat the latent tuberculosis infection, and the modified NPs showed a higher cell uptake in macrophages (97.2%) than unmodified ones (42.4%), thereby increasing intracellular antibiotic efficiency [172]. Other studies also reported mannosylated gelatin microspheres, gelatin NPs [242], and solid lipid NPs [243,244,245] to carry anti-tuberculosis drugs (such as INH and rifampicin) to target alveolar macrophages. In all cases, increased macrophage uptake and higher reduction in bacterial levels were observed compared with non-mannosylated particles, maintaining therapeutic concentrations for a prolonged period even upon the administration of a reduced clinical dose [170,242,246]. Despite of the mannose, Sharma et al. explored wheat germ agglutinin coated poly (lactide-co-glycolide) NPs as nanocarriers of rifampicin, INH, and pyrazinamide to treat tuberculosis. This nanosystem could reduce the frequency of antitubercular drug administration, therefore improving patient compliance with tuberculosis chemotherapy [247].

Leishmaniasis is a common tropical infectious disease characterized by fever, anemia, weight loss, and the enlargement of the spleen and liver [203]. The etiological agents of leishmaniasis are protozoan parasites called *Leishmania donovani*, which are intracellular parasites targeting mononuclear phagocytes (monocytes and macrophages) and replicating within membrane-bound subcellular organelles. The parasites develop several mechanisms to survive in macrophages and inhibit parasite-specific cell-mediated host immune response [248]. The classical treatment of leishmaniasis is not effective due to drug resistance, toxicity, bioavailability, and cost [249]. Current treatment for leishmaniasis mainly depends on amphotericin B (AmB), which also has limitations, such as dose-related hematologic toxicity [250], naphrotoxicity [251], stability, and high cost. Therefore, targeted intracellular delivery of AmB is required to enhance drug efficiency and facilitate pathogen clearance. As mentioned previously, mannose-based carriers can specifically target macrophages via interaction with mannose receptors on the cell surface, so studies have developed mannose-anchored thiolated chitosan AmB nanocarrier complexes (MTC AmB) for clearance of *Leishmania* in macrophages. The result showed a 71-fold increase in MTC AmB uptake compared with native drugs and a 13-fold enhancement in drug efficacy [252]. Despite the mannose, Singh used PhoS, which can be recognized by scavenger receptors (such as CD68 and CD14) on macrophage surfaces for macrophage-targeting. They fabricated PhoS anchored PLGA NPs to deliver AmB, and found that those NPs preferentially accumulated in macrophage-rich organs, which significantly increased anti-leishmanial activity and continually released the drug within 72 h [203].

Human immunodeficiency virus (HIV) infects approximately 35 million people globally and results in 1.8 million deaths every year [253,254]. HIV mainly targets and survives in macrophages, assembling and accumulating in intracellular compartments in macrophages, thus escaping from immune clearance [255]. Given that inhibition of HIV replication could enhance the host response and control infections, studies have been performed to develop macrophage-targeting nanosystems to deliver anti-HIV drugs [256,257]. Zhou et al. decorated NPs with folic acid (FA), which can target folate receptor-β expressed on the surface of activated macrophages [258]. Long-acting cabotegravir, which is an antiretroviral drug, was encapsulated in the nanoparticle and specifically delivered into macrophages. The result showed the slow release of drugs from macrophages, allowing for sustained intracellular drug levels, thus facilitating long-term viral suppression [258].

Among individuals infected by HIV, approximately one-third of them are co-infected with *Mtb*. HIV-1 infection causes severe damage to the immune system, which enables *Mtb* to infect and survive in the body more easily. Meanwhile, *Mtb* infection increases HIV-1 replication, thus enhancing the severity of HIV infection. Given that both HIV-1 virus and *Mtb* mainly reside in mononuclear macrophages, Narayanasamy and colleagues introduced macrophage-targeting long-acting gallium (Ga) nanoformulation for drug delivery. As a crucial element for the metabolism and growth of most microorganisms, including *Mtb* and HIV, Ga could stay inside the macrophages for a long time, exhibiting long-term antivirus effects [259].

*S. aureus* is a Gram-positive bacterium that predominantly infects the skin and the respiratory system, and how to treat *S. aureus* infection in deep tissue remains a major challenge. Local infection can process into the most serious systemic *S. aureus* infection-sepsis [260]. The current approach to treat *S. aureus* infection is using small molecule antibiotics, which cause side effects, such as toxicity and drug resistance [261]. Due to their crucial function in immune response, macrophages are potential targets for anti-infection. Kim synthesized porous Si NPs carrying siRNA against *Irf5* to promote phagocytosis of macrophages and inhibit their inflammatory function [262]. The porous Si NPs contain an outer sheath of homing peptides and fusogenic liposomes, allowing them to selectively target macrophages. *Irf5* gene highly exists in M1 macrophages, upregulating inflammatory factors while downregulating anti-inflammatory cytokines [263,264]. Knockdown of *Irf5* in the early stages of staphylococcal pneumonia can prevent the excretion of inflammatory cytokines and reverse prolonged inflammation, allowing the immune system to clear bacteria and repair tissue [78,265].

Nanotechnology has also been applied to develop novel vaccines against pathogens capable of inducing robust and protective autoimmunity. Chavez-Santoscoy et al. decorated the surface of polyanhydride NPs with specific carbohydrates to provide pathogen-like properties. The carbohydrates, galactose and di-mannose, which were found on the surface of respiratory pathogens, can facilitate both macrophage-targeting and immune activation [266,267,268,269]. This nanovaccine can promote robust pulmonary immunity against many respiratory pathogens, including *Yersinia pestis*, *Mtb*, *Streptococcus pneumoniae* and influenza viruses [178]. To develop an effective vaccine against HIV infection, a macrophage-targeting HIV immunotherapeutic vaccine based on NPs was introduced. The Ebola virus envelope glycoprotein was incorporated with non-replicating virus-like particles (VLPs) to enhance the NP’s macrophage and dendritic cells targeting capability, resulting in a stronger HIV-specific humoral immune response [270]. Hattori et al. introduced mannosylated liposomes, as DNA vaccine carriers and revealed enhanced Th1 immune response, suggesting that nanotechnology was a potent method for DNA vaccine therapy [1].

### 6.4. Macrophage-Targeting NPs for Bone Regeneration

Bone defects caused by diseases, such as trauma, tumor, and infection, often lead to the non-union of bones, delayed healing or non-healing, and local dysfunction. Although autologous bone transplantation has been widely used to cure bone defects, it has several disadvantages, such as the limited availability of graft volume, the morbidity of the donor site, and the prolonged operation time [271,272,273,274]. To solve these problems, synthesized biomaterials incorporated with osteoinductive factors have become a promising way to treat bone defects [271,272,273,274]. Recent studies have found that the immune system is closely associated with the skeletal system by regulating the biological behavior of bone cells [275,276,277,278], especially for biomaterial applications. Once a foreign material enters the body, it is immediately recognized by the immune system. It triggers activation/inflammation of the immune system, which then influences the subsequent bone regeneration and eventually determines the success of bone biomaterial application in vivo [277,278]. The relationship between foreign biomaterials, host immune cells, and bone cells, is termed as “osteoimmunomodulation”, and biomaterials with appropriate osteoimmunomodulatory capacity can, therefore, modulate local immune microenvironment to favor osteogenesis [23,279].

Among immune cells, the macrophage is one of the most important cell types. The upregulated release of proinflammatory cytokines, including IL-1β, IL-6, and TNF-α from M1 macrophages results in suppression of osteogenesis [280,281] and increased osteoclastogenesis [92]. On the contrary, the anti-inflammatory M2 phenotype can release osteogenic cytokines, such as BMP2 and VEGF, to eliminate inflammation and promote bone healing [20,282,283]. Therefore, nanomaterials are emerging as effective agents able to target macrophages, inducing their M2 polarization and thus modulating bone regeneration. One way to stimulate M2 polarization is to change the surface chemistry of NPs by using bioactive molecules, such as conjugating gold NPs with RGD [91], hexapeptides Cys-Leu-Pro-Phe-Phe-Asp [92], and IL-4 [93], or coating hydroxyapatite on the surface of CeO_2_ NPs [98]. It is noteworthy that some NPs themselves can enhance M2 polarization, such as gold, TiO_2_, and CeO_2_ NPs [94,95,97]. In addition, the nanopore structure and pore size were found to affect the spreading and cell shape of macrophages by modulating their adhesion, which subsequently influences their autophagy, inflammatory response, and release of osteogenic factors [284,285]. For example, Chen found that macrophages on surfaces with larger sized pores (100 and 200 nm) become more anti-inflammatory, producing more pro-inflammatory cytokines and increasing the production of M2 surface-markers [284]. Surface roughness of biomaterials also influences macrophage polarization and cytokine secretion. Studies indicated that titanium with the smooth surface could stimulate M1 macrophage activation, expressing inflammatory cytokines, such as IL-1β, IL-6, and TNF-α, while rough and hydrophilic titanium surface enhanced anti-inflammatory macrophage polarization with the increased secretion of IL-4 and IL-10 [286].

Using nanosystems as carriers to deliver bioactive molecules (including growth factors, cytokines, gene-modulators, and signaling pathway regulators) is another way to induce M1-to-M2 polarization of macrophages. For example, exogenous addition of sphingosine-1-phosphate (S1P), which is a sphingolipid growth factor, stimulates macrophages toward M2-like phenotype [287]. In the study by Das et al., fused nanofibers loaded with S1P synthetic analog were used to direct macrophage polarization towards M2-like phenotype in a mandibular bone defect model and successfully facilitated the osseous repair [288]. Yin et al. developed gold nanocages (AuNC) coated with LPS-stimulated macrophages cell membranes to deliver an anti-inflammatory drug named esolving D1. After LPS stimulation, cytokine receptors on the cell membrane were overexpressed and were able to neutralize pro-inflammatory cytokines [289]. This nanosystem was found to effectively reverse inflammation, facilitate M2 activation and promote osteogenesis in the femoral defect. IL-4 is a well-known anti-inflammatory cytokine, so various nanocarriers have been developed to load IL-4 to induce M2 polarization [290,291,292]. A nanofibrous heparin-modified gelatin microsphere incorporated with IL-4 was developed to resolve the chronic inflammation caused by diabetes and enhance osteogenesis [291]. In another study, the gene of CD163 (a M2 macrophage marker belonging to the scavenger receptor cysteine-rich family) was encapsulated into polyethyleneimine NPs decorated with a mannose ligand to selectively target macrophages to transfer them into anti-inflammatory phenotype [293].

### 6.5. Macrophage-Targeting NPs for Atherosclerosis

Atherosclerotic cardiovascular disease is a chronic inflammatory disease. During the progress of atherosclerosis, macrophages give rise to early lesions by engulfing low-density lipoprotein (LDL)-derived cholesterol to form lipid-laden foam cells, and the accumulation of cholesterol in macrophages promotes inflammatory responses, thus amplifying inflammation and leading the lesions to progress to an advanced stage [43]. The increasing apoptosis of macrophages leads to plaque necrosis, a key feature of ‘vulnerable’ plaques [44,46,294] characterized by a thin fibrous cap and a large necrotic core. Traditional ways to reduce the risk of cardiovascular events, such as lipid-lowering therapies and anti-inflammatory therapies, have limitations, including low compliance, low bioavailability, poor target specificity, and safety concerns. Therefore, nanotechnology has addressed these challenges and improved disease therapy [295,296].

The pathological processes of atherosclerosis include macrophage recruitment and proliferation; defective efferocytosis, which results in a defect of dead cells movement; plaque inflammation, and cholesterol and oxidized LDL accumulation. Nanotechnology can thus target these processes for atherosclerosis treatment [76]. For instance, monocytes adhere to the arterial wall through cell adhesion molecules, such as intercellular adhesion molecule 1 (ICAM1) and vascular cell adhesion molecule 1 (VCAM1). Sager et al. used polymeric NPs to deliver siRNA to silence five endothelial cell adhesion molecules and inhibited monocyte recruitment to atherosclerotic lesions [297]. Rapamycin reduces the levels of pro-inflammatory cytokines via inhibition of the NF-κB pathway, so targeted delivery of rapamycin by biomimetic NPs [298,299,300] was developed to suppress the proliferation of macrophages and reduce inflammation. The upregulation of CD47 (a “do not eat me” signal) in plaque macrophages can lead to efferocytosis deficiency, so anti-CD47 antibody therapy can efficiently reactivate efferocytosis [301]. Flores et al. encapsulated inhibitor of signal-regulatory protein-α (SIRPα), an antiphagocytic target of CD47, in single-walled carbon nanotubes (SWNTs) for macrophage-specific delivery, and restored efferocytosis in mice model [302]. To reduce plaque inflammation, anti-inflammatory therapeutics were extensively used. IL-10, a well-known anti-inflammatory cytokine, was loaded in PLGA-PEG diblock copolymer NPs and RGD peptide-conjugated pluronic-based nanocarriers to effectively target plaque and decrease inflammation [303,304]. Inhibitors of NF-kB signaling pathways, such as celastrol, and agonists of anti-inflammatory peroxisome proliferator-activated receptor gamma (PPARγ) signaling pathways, such as pioglitazone, were also carried by macrophage-targeting NPs for anti-inflammatory treatment of atherosclerosis [305,306]. A recent study by Wu used M2-like macrophage-derived exosomes as a nanocarrier to deliver anti-inflammatory bio-products. The nanocarrier itself can release anti-inflammatory cytokines, and together with the drug inside of it, the nanosystem promotes the regression of atherosclerotic plaques in Apoe−/− mice [307]. Cholesterol accumulation in plaque macrophages is associated closely with the severity of atherosclerosis. The macrophage ATP-binding cassette (ABC) transporters ABCA1 and ABCG1 are able to transport excess cholesterol to extracellular apolipoprotein A-I (apoA-I) and high-density lipoprotein (HDL), thereby increasing ABCA1 or ABCG1 expression can promote cholesterol efflux from macrophages [308,309,310,311]. miR-33 inhibits cholesterol efflux while miR-206 and miR-223 increase cholesterol efflux, so miR-206, miR-223, and an antisense oligonucleotide against miR-33 were encapsuled in macrophage-targeting NPs to induce cholesterol efflux and reduce plaque burden. The macrophage-targeting ability of these NPs was afforded by cyclic pentapeptides cyclo(-Arg-Gly-Asp-D-Phe-Lys-)(cRGDfK) for integrin receptors on macrophages [312,313]. Despite of miRNA, activation of sterol-regulated transcription factor liver X receptor-α (LXRα) also upregulates express of ABCA1 and ABCG1. Yu and colleagues introduced LXR agonist GW3965 conjugated collagen type IV-targeted polymeric NPs and effectively promoted the cholesterol efflux [313].

### 6.6. Macrophage-Targeting NPs for Other Inflammation-Related Diseases

A surgical suture is a medical approach to closing the wound of skin and organs, whereas excessive inflammation surrounding the suture can disrupt the wound healing process. To solve this problem, poly(lactic-co-glycolic) NPs decorated with macrophage-targeting ligand mannose were developed to deliver diclofenac, which is an anti-inflammatory drug, as a local anti-inflammatory device. The nanosystem showed an enhanced anti-inflammatory effect in the excisional wound healing animal model compared to free drug-coated suture [314].

Acute lung injury (ALI) is an acute and life-threatening pulmonary inflammatory disease characterized by acute hypoxic respiratory failure caused by pulmonary and nonpulmonary insults [315,316,317]. The pathology of ALI includes infiltration of inflammatory cells, an increase in cytokine release, and uncontrolled inflammation, which further contributes to the accumulation of proteinaceous edema in pulmonary tissue, causing overall acute, diffuse, and inflammatory lung injuries [318]. So far, low tidal volume mechanical ventilation is the only effective therapy of ALI [319,320], but the mortality remains high. Therefore, new therapeutic approaches are needed to limit the acute lung inflammation and induce tissue repair. Macrophages play a crucial role in the inflammation of ALI and, therefore, are considered as a therapeutic target of ALI intervention [321,322]. As mentioned before, gold NPs coated with hexapeptides on the surface could stimulate the M2 restorative polarization of macrophages by inhibiting TLR signaling in macrophages [323], so Wang applied this nanoparticle in LPS induced ALI mouse model to reduce inflammatory cell infiltration, increased M2 polarization and alleviated lung inflammation [92].

Rheumatoid arthritis is an autoimmune disease which tightly related with the activation of macrophages in arthritic joints [324]. The release of inflammatory cytokines induces a rapid nuclear translocation of NF-kB, which then interacts with response-associated genes (such as TNF-α), and activates their transcription [325]. As such, blocking the NF-kB signaling pathway could reverse the inflammation, thus curing autoimmune diseases, such as rheumatoid arthritis. Hattori investigated the delivery of NF-kB-decoy by using folate-linked lipid-based NPs, which selectively target activated macrophages. The decoy was directed at a cognate sequence of NF-kB using a double-stranded oligonucleotide. It blocked the intracellular signaling pathways, resulting in inhibited translocation of NF-kB into the nucleus and decreased levels of inflammatory cytokines [326].

Obesity alters adipose tissue metabolic and endocrine function, which leads to an increased release of fatty acids, hormones, and proinflammatory molecules that contribute to obesity-associated complications [327]. Recent investigations suggest that obesity gives rise to a state of chronic, low-grade inflammation, contributed to by adipose tissue macrophages (ATM) [328,329]. In animal experimentation, lean mice were found to express more M2 phenotype-associated genes, while obese mice expressed more M1 macrophages genes, suggesting that obesity stimulates a M2-to-M1 phenotype change in ATM [330]. In many studies, nanotechnology has been used to delivery anti-obesity therapeutics to increase the stability, solubility, and bioavailability of therapeutics, protecting them from fast degradation in the body, thus prolonging their circulation time [331,332,333]. Moreover, given the crucial role of macrophages in obesity, using NPs to target ATMs in adipose tissue and modulate their polarization may be a direction for anti-obesity treatment. Zhao et al. investigated the function of celastrol-loaded nanomicelles in diet-induced obese mice. The nanosystem significantly reduced the expression of macrophage M1 biomarkers and increased the expression of macrophage M2 biomarkers in a dose-dependent manner, representing a translatable therapeutic opportunity for treating diet-induced obesity in humans. To selectively target ATM and reduce the off-target of small-molecule drugs, Ma invented nanoscale polysaccharides conjugated with dextran, a biocompatible glucosepolymer which can efficiently target macrophages. They loaded anti-inflammatory drugs in the nanoparticle for therapeutic modulation of macrophage phenotype and significantly reduced pro-inflammatory markers in adipose tissue of obese mice, providing promising nanomaterials-based delivery strategy to inhibit obesity [334].

The word “inflammageing” describes a chronic low-grade inflammation observed in ageing population, characterized by an elevated level of circulating inflammatory mediators, such as C reactive protein (CRP), IL-6, IL-1β, and TNF-α [335,336]. Recent data have suggested that although macrophages are not the only source of inflammageing, they are a central component in initiating the phenomenon [335]. Therefore, anti-inflammatory treatment targeting at macrophages may reverse age-related chronic inflammation, thus ameliorating the decline of physiological functions in ageing [337]. Arnardottir et al. applied novel nano-proresolving medicines carrying anti-inflammatory molecules (resolvins D1 and D3) in aged mice and observed promoted M2 macrophage polarization, as well as reduced exacerbated inflammatory response [338]. In our research, we developed an *Escherichia. coli* OMV coated AuNCs carrying anti-inflammatory drugs to reverse age-related inflammation. Since M1 macrophages are microbicidal, OMVs are supposed to be specially recognized by receptors on M1 macrophages membrane, thus achieving selectively M1 macrophage targeting. Although macrophage-targeting drug delivery nanotechnology has not been largely applied in inflammageing, it provides a new horizon for improving age-related diseases characterized by excessive innate inflammatory responses.

## 7. Conclusions and Future Direction

Cell-targeting nanotechnology is extensively investigated in a wide range of inflammation-related diseases. This novel approach successfully improves drug efficiency and reduces off-target therapeutics, thus lessening the toxicity of conventional small molecule drugs. Given the vital role of macrophages in immune response and pathology of various diseases, macrophage-targeting NPs have been reported to modulate macrophage behaviors, thus achieving more accurate diagnosis and treatment effects of diseases. NPs can either passively target macrophages by preferentially accumulating in inflammation sites and be taken up by phagocytes, or positively target macrophages by specific interaction between agents decorated on NP surfaces and receptors on macrophages membranes that recognize these agents. This review has summarized current approaches to developing macrophage-targeting NPs (including both passive and positive ways), and their applications in diagnosis and therapy. Despite the numerous advantages of NP-based therapy, there are still issues to be addressed for the optimization of NP treatment. Firstly, targeting M1/M2 macrophages is a challenge, as macrophage phenotype occupies a continuum between M1 and M2 designations, which means macrophages usually possess surface markers of both phenotypes, making it difficult to distinguish M1 from M2 macrophages. Thus, how to detect macrophage populations relying on markers remains to be problematic. Second, the most proper time to intervene the macrophage polarization still needs to be further explored. For example, in bone regeneration, the M1 phenotype dominates the early stage of inflammation, while M2 macrophages play a prominent role in the later stage. A timely M1–M2 switch determines the appropriate immune environment that decides later bone remodeling. However, the proper time to complete this transformation remains unknown and needs to be discovered in the future. Thirdly, although macrophage-targeting NPs have achieved the spatial control of drug release in the targeted cells, the temporal control of drug release still needs to be improved. Most NPs can enable a slow release of loaded therapeutics, avoiding the robust increase in local drug concentration, thereby reducing side effects and enhancing the efficiency of drugs. Some NPs are able to more precisely achieve temporal control of drug release, such as gold nanocages, which are supposed to release drugs only after near-infrared radiation [339]. However, drug leakage still exists before the radiation due to the slow melting of phase-change materials [340]. Therefore, novel NPs with reduced drug leakage and more accurate temporally release control need to be developed. Moreover, though modulation of the immune profile of macrophages by NPs has shown great promise in animal experiments, their efficacy in humans cannot be guaranteed due to the differences in pathobiology between animal models and humans. For example, the procession of atherosclerosis is usually faster in animal models compared with that in humans, suggesting a more complicated and heterogeneous pathology in the human body [341]. In addition, macrophage subtypes also vary in different individuals, making personalized nanomedicine particularly crucial to achieving the desired therapeutic outcomes [76]. In addition, the physicochemical interactions between NPs and targeted cell surfaces, the pharmacokinetics and biodistribution of NPs, and the underlying mechanism of their immunomodulation need further investigation to provide the foundation for improving nanotherapeutics in the future.

## Figures and Tables

**Figure 1 biomedicines-10-01203-f001:**
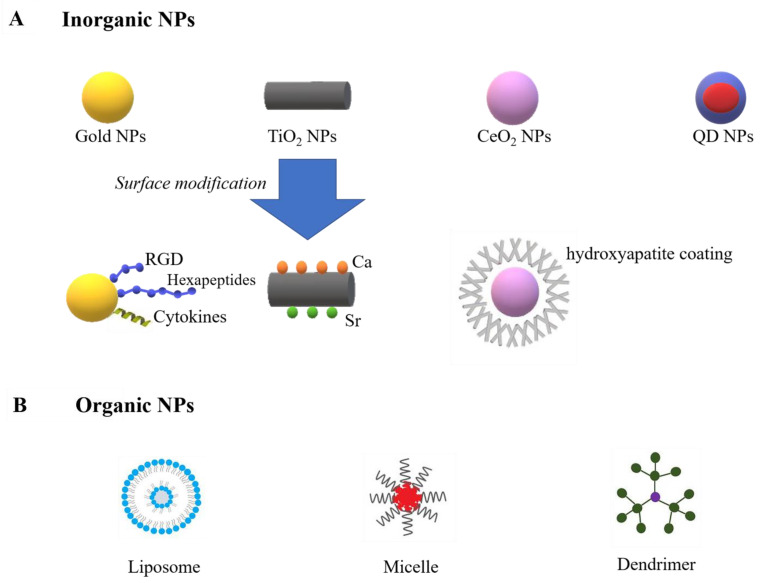
Current nanomaterials for regulating macrophage polarization. (**A**) Surface modification of inorganic NPs to modulate macrophage polarization. (**B**) Organic NPs.

**Figure 2 biomedicines-10-01203-f002:**
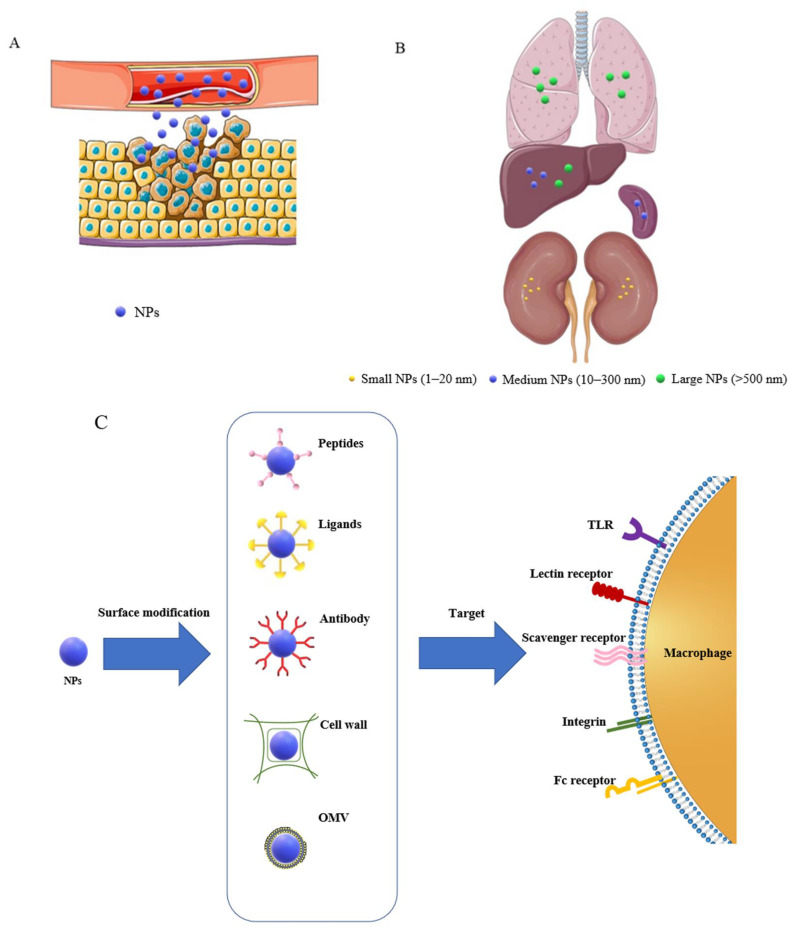
Summary of macrophage-targeting nanotechnologies. (**A**) Passive macrophage-targeting. NPs accumulate in tissues through the vascular leakage. (**B**) Passive macrophage-targeting. NPs with different sizes preferentially accumulate in different organs. (**C**) Active macrophage-targeting. With various surface modification methods, NPs can specially target macrophages via recognition by receptors on macrophage membrane.

## Data Availability

Not applicable.

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
