# Peer review of "Current Development of Nano-Drug Delivery to Target Macrophages"

_biomedicines, 2022, doi:10.3390/biomedicines10051203_

Round 1

Reviewer 1 Report

The review article entitled „Current development of nano-drug delivery to target macrophages” by Cai, D. et al. describes various nanostructures-based strategies for modification of macrophages. The manuscript is well organized but due accumulation of large quantities of information it is difficult to read in places. It consists of three distinct parts; one describes the role of macrophages in physiology and pathophysiology. Second part shows how nanostructures interact in immunological system in general and macrophages in particulars. Third part describes large number of specific nanostructure designs to target macrophages in disease treatment. The article design provides sufficient information to guide reader through such difficult and broad topic. The proposed approach used to organize information is sufficient, but the last part of the manuscript is overwhelming.

In summary, assuming that structure of the article is correct, it is practically ready for publication.

I have only two minor comments;  

  • due to the large number of abbreviations, which are used in different places, it would be helpful add the list of abbreviations,
  • line 270, the statement that “liposomes are large NPs (>1000 nm)” is not correct.

Author Response

Response to Reviewer 1 Comments

Comment: The review article entitled“Current development of nano-drug delivery to target macrophages” by Cai, D. et al. describes various nanostructures-based strategies for modification of macrophages. The manuscript is well organized but due accumulation of large quantities of information it is difficult to read in places. It consists of three distinct parts; one describes the role of macrophages in physiology and pathophysiology. Second part shows how nanostructures interact in immunological system in general and macrophages in particulars. Third part describes large number of specific nanostructure designs to target macrophages in disease treatment. The article design provides sufficient information to guide reader through such difficult and broad topic. The proposed approach used to organize information is sufficient, but the last part of the manuscript is overwhelming.

In summary, assuming that structure of the article is correct, it is practically ready for publication.

Response: We thank the reviewer for your positive comments. The structure of this article is rationally designed and explained as follows: we firstly introduced the role of macrophages in pathological process of inflammatory diseases, and showed why macrophage-targeting is necessary in disease treatment. Then we introduced nanoparticles as potentials tool to regulate macrophage responses, by explaining how nanoparticles interact with immune system and presented their advantages as drug carriers for macrophage-targeting. These three parts provide fundamental information for the discussion of current macrophage-targeting nanotechnologies and their potential therapeutic applications. We believe the drafted structure is logically designed to help readers to have a better understanding of this topic.

I have only two minor comments; 

Point 1. Due to the large number of abbreviations, which are used in different places, it would be helpful add the list of abbreviations. 

Response 1:We thank for the valuable suggestion. The list of abbreviations has been added to the end of this manuscript.

Point 2.  Line 270, the statement that “liposomes are large NPs (>1000 nm)” is not correct.

Response 2: We thank the reviewer for pointing this out. We have corrected this description as the size of liposomes are between 25 and1000 nm.

Reviewer 2 Report

Authors:

The topic of the review on Current development of nano-drug delivery to target macrophages seems to be a review in the important field of science. The manuscript is more or less logically constructed. However, the manuscript is written in a bad way from the technical point of view. It contains numerous typing errors, missing Greek characters, bad formating of chemical formulae, bad formating of several paragraphs, errors in the chemical names and metal isotope labeling etc. The text placed inside the figures, namely that in the Figure 2, is very small for reading, the figures should be edited. Some references are incomplete, lacking journal names, pages etc.

Based on all these findings, I recommend to reject this manuscript.

Author Response

Response: We thank the reviewer for the critique on the structure, editing and misspelling issues. The revised manuscript has been carefully reviewed, restructured accordingly, and all the mistakes mentioned, including typing errors (eg. line 57,127, et al), missing Greek characters (eg. line 53,125, et al), formatting of chemical formulae (eg. line 248, 252, 254, et al.), formatting of several paragraphs (eg. line 380, et al), errors in the chemical names and metal isotope labeling (eg. line 444, 580, et al) etc., have been corrected accordingly. The Figure 1 and 2 have been edited to make sure the texts placed inside are legible to read. All the references have been checked and updated. We believe that after the revision, the quality of this manuscript has been significantly improved for your re-consideration of publication in this journal.

Round 2

Reviewer 2 Report

Authors:

The manuscript has been improved. I believe it can be accepted and published.